# POIVRE: SELF-REFINING VISUAL POINTING WITH REINFORCEMENT LEARNING

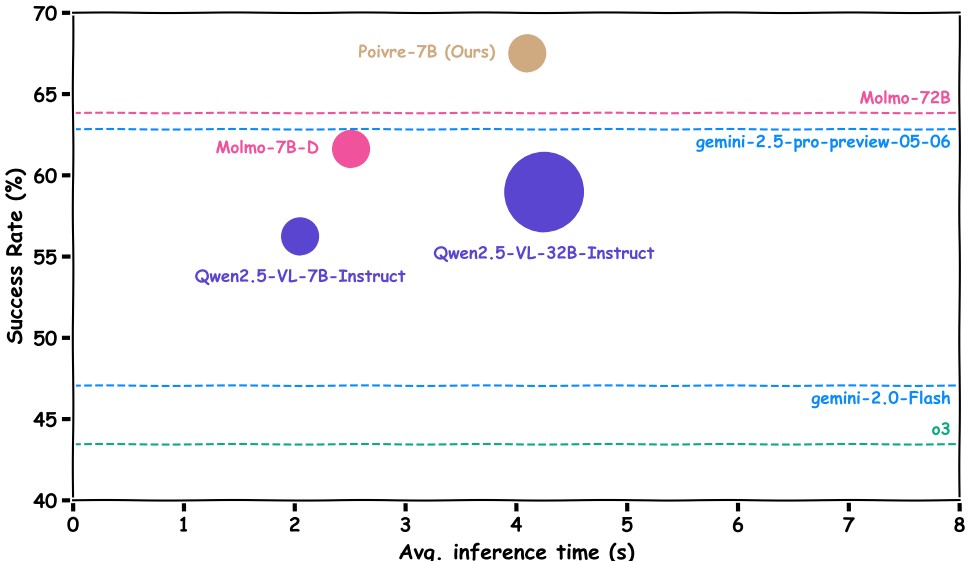

Figure 1: Performance of VLMs on Point-Bench (Cheng et al., 2025). Circle radius indicates model size. Dotted lines represent models whose average inference time is not available.

## ABSTRACT

Visual pointing, which aims to localize a target by predicting its coordinates on an image, has emerged as an important problem in the realm of vision-language models (VLMs). Despite its broad applicability, recent benchmarks show that current VLMs still fall far behind human performance on this task. A key limitation is that VLMs are typically required to complete the pointing task in a single step, akin to asking humans to point at an object without seeing their own fingers. To address this issue, we propose a simple yet effective self-refining procedure: ***Point, Visualize, then Refine*** (Poivre[1]). This procedure enables a VLM to first mark its estimated point, then iteratively refine the coordinates if necessary. Inspired by advances of reasoning models in the natural language domain, we employ reinforcement learning (RL) to incentivize this self-refining ability. For the RL training, we design a neat process reward that is not only empirically effective but also grounded in appealing properties. Our trained model, *Poivre-7B*, sets a new state of the art on Point-Bench, outperforming both proprietary models such as Gemini-2.5-Pro and large open-source models such as Molmo-72B by over 3%. To support future research, we release our training and inference code, dataset, and the Poivre-7B checkpoint.

---

[1]Poivre means "pepper" in French.

# 1 INTRODUCTION

Pointing is one of the most fundamental and intuitive mechanisms for grounding language in visual contexts. From early childhood communication to robotics and human–computer interaction, pointing provides a precise yet low-bandwidth way to bridge abstract intent and concrete spatial reference (Cheng et al., 2025). In the realm of VLM, visual pointing is typically formulated as predicting the coordinates of a target object in an image, given a natural language instruction. This capability is critical for a wide range of applications, including assistive technologies for the visually impaired (Yuan et al., 2024b), interactive educational tools (Hu et al., 2024), and robotic manipulation systems that require fine-grained spatial reasoning (Yuan et al., 2024a).

Despite its importance, recent benchmarks such as Point-Bench (Cheng et al., 2025) reveal that even the most advanced VLMs still fall far short of human-level performance. While humans naturally refine their gestures by observing and adjusting, current VLMs are usually constrained to produce a pointing result in a single step. This mismatch leads to inaccurate predictions and limits the robustness of VLMs in realistic settings. Analogous to asking humans to point without seeing their own fingers, the one-shot paradigm overlooks the natural process of iterative refinement that underlies human pointing behavior.

To address this gap, we propose *Point, Visualize, then **Re**fine* (Poivre), a simple yet effective procedure that enables self-refining for visual pointing. Under this procedure, a model first generates an initial estimation of the target location, then visualizes this prediction by marking it on the image, and refines its estimate in subsequent rounds if necessary. This iterative interaction not only makes the task more natural and improve robustness, but also opens a path for VLMs to generalize beyond their training setup by extrapolating to more refinement steps at inference time.

Inspired by advances in reasoning models within the natural language domain, we employ reinforcement learning to incentivize this self-refining ability. In particular, we introduce a process reward inspired by potential-based reward shaping (PBRS), which encourages the model to improve across refinement steps rather than optimizing only for the final outcome. Our RL-trained model, Poivre-7B, achieves new state-of-the-art results on Point-Bench (Cheng et al., 2025), surpassing both proprietary models (e.g., Gemini-2.5-Pro) and large open-source models (e.g., Molmo-72B). Moreover, Poivre-7B demonstrates strong generalization on robotics benchmarks such as where2place from RoboPoint (Yuan et al., 2024a), despite not being specifically trained on robotics datasets. Our contributions are summarized as follows:

- We propose the *Point, Visualize, then Refine* procedure, which allows VLMs to iteratively refine predictions by observing their own outputs, making the pointing task more natural and improve robustness compared to the one-shot setting.

- We design a PBRS-inspired process reward that encourages consistent improvement across refinement steps, extending beyond the conventional outcome reward. This leads to more effective RL training for visual pointing. Our RL-trained model, Poivre-7B, achieves a new state of the art, outperforming both proprietary models (e.g., Gemini-2.5-Pro) and large open-source models (e.g., Molmo-72B) by a significant margin.

- Beyond the results on Point-Bench, we further conduct experiments to demonstrate the effectiveness of our process reward, the extrapolation capability of the trained model, and its generalization to the robotics domain. Our findings indicate several promising future directions for research in visual test-time scaling and chain-of-thought prompting. To facilitate future research, we release our training and inference code, dataset, and the Poivre-7B checkpoint.

# 2 RELATED WORK

**Visual pointing.** Visual pointing has recently emerged as an important research problem in the VLM community. Molmo and Pixmo (Deitke et al., 2025) pioneered this direction by introducing open-source VLMs and datasets explicitly targeting the pointing task. In particular, the Pixmo-Points dataset contains 223k images paired with 2.3M question–point annotations, providing large-scale supervision for training visual pointing models. Following its release, several open-source models, including Qwen2.5-VL (Bai et al., 2025), incorporated pointing supervision to enhance their

performance. Beyond general-purpose VLMs, visual pointing has also been studied in robotics. For instance, RoboPoint (Yuan et al., 2024a) links natural language instructions with keypoints relevant for manipulation, bridging pointing with real-world control. PIVOT (Nasiriany et al., 2024) is a training-free method for robotic that iteratively prompting VLMs to select the best ones for the task. MOKA (Fang et al., 2024) is an approach that employs VLMs to solve robotic manipulation tasks specified by free-form language instructions. SoM (Yang et al., 2023) uses off-the-shelf interactive segmentation models as tools to answer the questions that require visual grounding. However, in contrast to the natural language domain, the application of reinforcement learning to visual pointing remains surprisingly limited, which motivates us to examine its potential for this task.

**Reinforcement learning for (V)LMs.** Reinforcement learning has been widely explored as a mechanism to align both language models and vision–language models with desired behaviors. In the language domain, RL has demonstrated remarkable success in reasoning-intensive tasks, such as mathematical problem solving (Shao et al., 2024; Luo et al., 2025; Xu et al., 2025). In multimodal contexts, most RL efforts have concentrated on visual question answering (VQA), often under the paradigm of "thinking with images" (Zheng et al., 2025b; Liu et al., 2025; Zhu et al., 2025). To the best of our knowledge, the most relevant work is VisionReasoner (Liu et al., 2025), which employs an L1 reward based on grounding points. While VisionReasoner achieves impressive results, it relies solely on single-turn RL, which limits its ability to incentivize self-refinement, a key focus of our work. Another notable study is Point-RFT (Ni et al., 2025), which leverages the pointing capabilities of VLMs to improve question-answering performance. However, Point-RFT addresses a fundamentally different task from ours. We also wish to highlight several excellent studies that explore a related but distinct task: graphical user interface grounding (Tang et al., 2025; Yuan et al., 2025; Zhou et al., 2025). While these works are highly impactful within their respective domains, to the best of our knowledge, none of them adopt the concept of self-refining VLMs as proposed in our work.

**Test-time scaling/Chain-of-thought.** Test-time scaling (TTS) has emerged as an active area of research within the language modeling community. Since the release of DeepSeek-R1 (DeepSeek-AI et al., 2025), numerous approaches have been proposed to enhance performance by extending inference beyond a single forward pass (Muennighoff et al., 2025; Ye et al., 2025). Of particular relevance to our work is the concept of iterative self-correction, where models refine their intermediate outputs to progressively improve reasoning quality (Kumar et al., 2025). Building on these ideas, we investigate self-refinement for visual pointing, enabling VLMs to iteratively refine their predictions through multiple rounds of feedback. Our approach can be viewed as a form of test-time scaling or visual chain-of-thought (COT) reasoning. However, further research is needed to bring visual self-refinement to the same level of maturity as its counterpart in the natural language domain.

## 3 PRELIMINARY

In this section, we introduce the visual pointing task and the RL algorithm adopted in this work.

**Problem statement.** We study the task of visual pointing with VLMs. Specifically, given an image $I \in \mathbb{R}^{H \times W \times 3}$ and a natural language description $q$ of one or more target objects, the VLM $\mathcal{F}_\theta$ generates a response $\mathcal{F}_\theta(I, q)$ from which we extract the predicted coordinates $P = \{(x_i, y_i)\}_{i=1}^K$. Our setting strictly follows Point-Bench (Cheng et al., 2025), we refer to that work for details.

**Reinforcement learning.** Unless otherwise specified, we adopt GRPO (Shao et al., 2024) as our default RL algorithm. We acknowledge several recent advances in RL algorithms, such as DAPO (Yu et al., 2025), GSPO (Zheng et al., 2025a), and GMPO (Zhao et al., 2025). Incorporating these methods in place of GRPO could potentially yield further improvements, but we leave this exploration to future work due to resource constraints. For each input $(I, q)$, GRPO samples $G$ rollouts $\mathbf{o} = \{o_1, o_2, \cdots, o_G\}$ from the old policy $\pi_{\theta_{\text{old}}}$ and optimizes the model by maximizing the following objective:

$$\mathcal{J}(\theta) = \mathbb{E}\left[\frac{1}{G}\sum_{i=1}^{G}\frac{1}{|o_i|}\sum_{t=1}^{|o_i|}\left\{\min\left[r_t(\theta)\hat{A}_{i,t}, \text{clip}\left(r_t(\theta), 1-\epsilon, 1+\epsilon\right)\hat{A}_{i,t}\right] - \beta\mathbb{D}_{\text{KL}}[\pi_\theta||\pi_{\theta_{\text{ref}}}]\right\}\right],$$

$$(1)$$

where $r_t(\theta) = \frac{\pi_\theta(o_{i,t}|(I,q),o_{i,<t})}{\pi_{\theta_{\text{old}}}(o_{i,t}|(I,q),o_{i,<t})}$ is the ratio function, $\hat{A}_{i,t} = \frac{r_i-\text{mean}(\mathbf{r})}{\text{std}(\mathbf{r})}$ is the advantage that is computed with the rewards $\mathbf{r} = \{r_1, r_2, \cdots, r_G\}$, $\epsilon$ and $\beta$ are hyperparameters, and $\mathbb{D}_{\text{KL}}$ is the unbiased estimator of the KL divergence.

Two parts of GRPO are especially important in our context: *rollout sampling* and *reward computation*. First, the input format for *rollout sampling* needs to match the one at inference time. This alignment encourages the model to learn the desired self-refining pattern. Second, the *reward computation* acts as our definition of a high-quality rollout, by making it clear which parts of the self-refining process matter most. Therefore, the reward function must be meticulously designed to accurately assess rollout quality and prevent reward hacking. We describe the detailed design of these two components in the following section.

## 4 METHOD

In this section, we detail the two core components in our RL training: *rollout sampling* and *reward computation*. The rollout sampling is structured to mirror the inference phase, strictly following our proposed *Point, Visualize, then Refine* procedure. For reward computation, we begin with a simple outcome reward and enhance it into a process reward by incorporating potential-based reward shaping (PBRS). We further show that this PBRS-inspired process reward offers advantages that are particularly suited to our task.

**Rollout/Inference.** Current VLMs are typically trained to accomplish the pointing task in a single step (Deitke et al., 2025; Bai et al., 2025; Cheng et al., 2025). Analogous to human behavior, this is akin to asking an individual to point at an object without being able to observe their own finger. To make the pointing task more natural and improve robustness, we introduce the ***Point, Visualize, then Refine*** (Poivre) procedure. The illustration of such procedure is presented in Figure 2. Poivre allows the model to first generate an initial estimation, visualize its own output, and refine the prediction, thereby mimicking how humans adjust their pointing after observing their own gesture. Formally, for a pointing task $(I_0, q)$, the VLM first predicts the coordinates of the target as $P_1$. The environment then visualizes this prediction on the original image to produce $I_1$. The image with marked points $I_1$ is then fed back to the model, allowing for further refinement. This iterative procedure naturally generalizes to any desired number of rounds $T$:

$$P_{i+1} = \mathcal{F}_\theta(\{I_j\}_{j=0}^i, q), \forall i \in \{0, 1, \cdots, T-1\}. \tag{2}$$

We consistently apply the Poivre procedure for both rollout sampling during RL training and inference. Experimental results in §5 show that the RL training effectively incentivizes the self-refining ability of the VLM.

**Reward.** For the visual pointing task, a straightforward reward function measures the distance between predictions and the ground truth. Given the final prediction $P_T$, a vanilla outcome reward is defined as:

$$R_{\text{O}}(d_T) = \exp(-\frac{d_T^2}{\sigma}), \tag{3}$$

where $\sigma$ is a hyperparameter and $d_T$ denotes the Euclidean distance from $P_T$ to the ground truth. This Gaussian-shaped function is intuitive, as it naturally captures diminishing marginal improvements when $d_T$ is small. However, relying solely on the outcome reward $R_{\text{O}}$ is suboptimal for the Poivre procedure, as it ignores intermediate predictions $\{P_j\}_{j=1}^{T-1}$. For example, a rollout that makes random guesses during the first $T-1$ steps but happens to produce a reasonable prediction in the final step receives the same reward as one that progressively refines its predictions. While this may seem harmless at first glance, such behavior is not ideal for RL training.

Inspired by potential-based reward shaping (PBRS) (Ng et al., 1999), we introduce a process reward that leverages the *difference of potentials*: $F(s, a, s') = \Phi(s') - \Phi(s)$, where $s$ and $s'$ are states, $a$ is an action, and $\Phi$ is a potential function over states. In our setting, the states correspond to the distances to the target, $d_t$ and $d_{t+1}$, the action is the new prediction $P_{t+1}$, and the potential function $\Phi(d_t)$ is naturally defined as $R_{\text{O}}(d_t)$. Building on this formulation, we propose the following PBRS-inspired process reward:

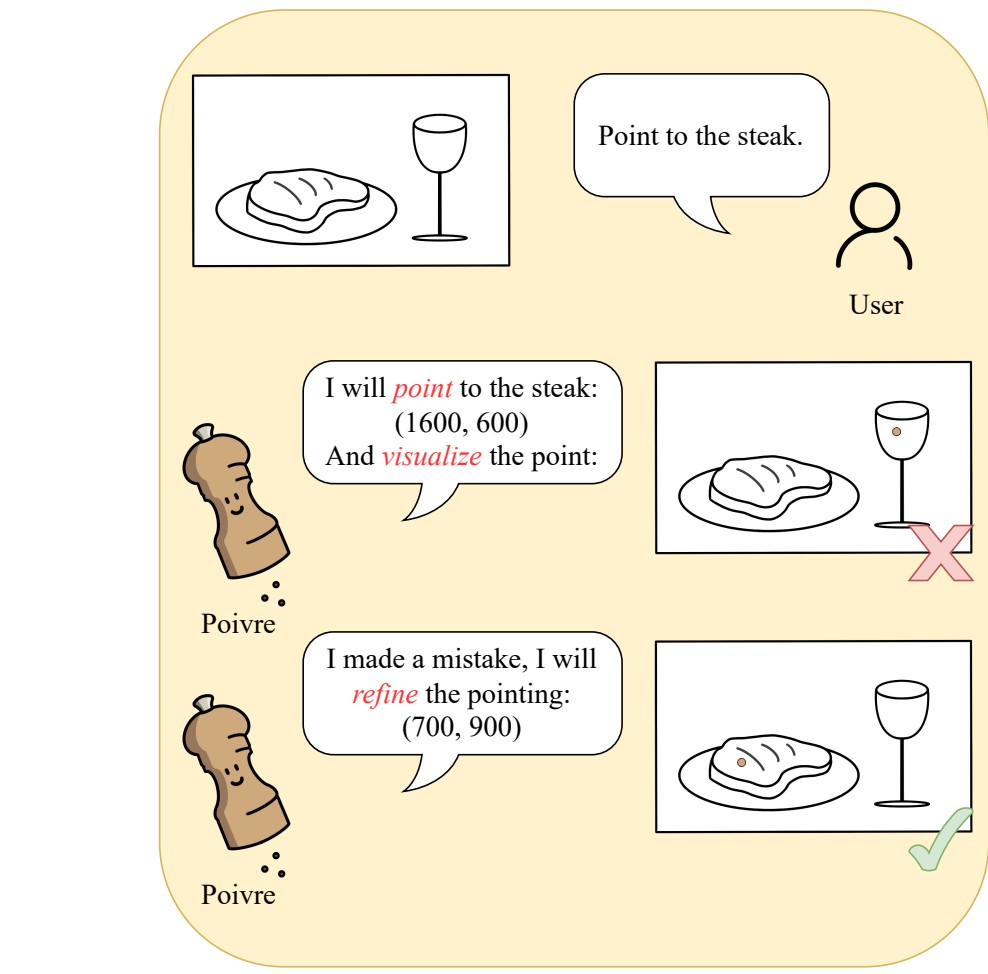

Figure 2: Illustration of our *Point, Visualize, then Refine* procedure. Brown markers denote the visualized coordinates. The cross and tick symbols are shown only for illustration purposes, no external verification is available to the models during rollout/inference.

$$R_{\mathrm{P}}(d_T) = \underbrace{R_O(d_1)}_{\text{Pointing reward}} + \sum_{j=1}^{T-1} \gamma^j \underbrace{(R_O(d_{j+1}) - R_O(d_j))}_{\text{Refinement reward}}, \quad (4)$$

where $\gamma \in (0, 1)$ is the discount factor. The process reward can be intuitively decomposed into two components: the *pointing reward*, which evaluates the initial prediction, and the *refinement reward*, which measures the quality of subsequent refinements. Experimental results in §5 show the advantage of our PBRS-inspired process reward over the vanilla outcome reward.

Beyond the empirical results, we also emphasize certain properties that help elucidate the advantages of the PBRS-inspired process reward. Proposition 4.1 demonstrates that this reward can also be interpreted as a weighted average of the pointing rewards across turns. The proof is deferred to the appendix.

**Proposition 4.1** *The PBRS-inspired process reward can also be viewed as the weighted average of the pointing rewards across turns, where the initial and the final estimation are most important for usual parameter choices. Formally,*

$$R_{\mathrm{P}}(d_T) = \gamma^{T-1} R_O(d_T) + \sum_{j=1}^{T-1} \gamma^{j-1}(1-\gamma) R_O(d_j), \quad (5)$$

*where* $\gamma^{T-1} \geq 1 - \gamma \geq \gamma^{j-1}(1 - \gamma), \forall j \in [T-1]$, *if* $T \leq 1 + \frac{\log(1-\gamma)}{\log(\gamma)}$.

Proposition 4.1 indicates that the PBRS-inspired process reward, when interpreted as a weighted average across turns, places greater emphasis on the first and last rounds. This weighting is intuitive, as the initial estimation determines the difficulty of the refinement process, while the final prediction reflects the ultimate quality.

## 5 EXPERIMENTS

This section details our experimental setup and presents the results. Our evaluation is designed to answer the following research questions:

- *RQ1:* How effective is Poivre on the visual pointing benchmark?
- *RQ2:* Does our PBRS-inspired process reward lead to performance gains compared to a simple outcome reward?
- *RQ3:* Can Poivre extrapolate to a greater number of refinement rounds at test time than were used during training?
- *RQ4:* Does the pointing ability of Poivre generalize to other application domains, such as robotics?
- *RQ5:* Is multi-turn RL necessary? Can the vanilla RL be employed to incentivize self-refinement, as has been demonstrated in the natural language domain?
- *RQ6:* What does the Poivre procedure look like on real-world images?

### 5.1 EXPERIMENTAL SETUP

**Training data.** We use the publicly available Pixmo-Points dataset (Deitke et al., 2025), which was specifically collected for the visual pointing task and contains 223k images paired with 2.3M question–point annotations. To adapt the dataset for RL training, we filter out corrupted samples and overly long instances, and then subsample 8,192 pairs due to resource constraints. The resulting dataset, referred to as *Pixmo-Points-RL-8K*, has been publicly released. We are aware that both the scale and the scope of training data can be further expanded, which we will leave for future work.

**RL hyperparameters.** We initialize training from the Qwen2.5-VL-7B-Instruct checkpoint (Bai et al., 2025). The batch size is set to 256, with 8 rollouts per sample, and a KL coefficient of 0.01. Notably, the number of scaling rounds $T$ is fixed at 2 during RL training, but this setup encourages the model to extrapolate beyond $T$ at test time. The resulting behavior will be presented later. The $\sigma$ in the PBRS-inspired process reward is set to 10 due to the normalization of the coordinates. Hyperparameters are fixed for all types of RL training.

**Resources.** RL training is conducted on 4 nodes, each equipped with 32 CPUs and 8 A100 GPUs. A single RL run takes approximately 16 hours, which corresponds to an estimated cost of around $2,000 on standard rented servers.

**Reproducibility statement.** We release both the training and inference code at `https://anonymous.4open.science/r/ICLR2026-Poivre`. The training dataset and the trained model, *Poivre-7B*, are publicly available on HuggingFace. The corresponding link will be provided after the double-blind review phase.

### 5.2 EXPERIMENTAL RESULTS

**Main results. (RQ1)** We denote our RL-trained model as *Poivre-7B*. We benchmark it against a wide range of competitive baselines, including the Molmo series (Deitke et al., 2025), Gemini series (Google, 2025), OpenAI series (OpenAI, 2024), Claude-3.7-Sonnet (Anthropic, 2024), Grok-2-Vision (xAI, 2024), LLaVA series (Li et al., 2024), Qwen series (Bai et al., 2025), and VisionReasoner-7B (Liu et al., 2025). We also include the most recent Qwen3-VL model, for which a technical report is not yet available at the time of writing. As reported in Table 1, Poivre-7B sets a new state of the art on Point-Bench (Cheng et al., 2025), surpassing the strongest baseline by a clear

Table 1: Success rates of VLMs on Point-Bench. The best results are **bolded**. The runner-up results are underlined. Performance of baselines are obtained from the Point-Bench paper if available.

|  | Success Rate |
| --- | --- |
| Human | 89.128 |
| grok-2-vision-latest | 20.530 |
| claude-3-7-sonnet-20250219 | 22.222 |
| GPT-4o | 29.502 |
| GPT-4.1 | 33.256 |
| o3 | 43.446 |
| gemini-2.5-flash-preview-04-17 | 46.960 |
| gemini-2.0-Flash | 47.052 |
| gemini-2.5-pro-preview-05-06 | 62.830 |
| llava-onevision-qwen2-7b-ov-hf | 6.324 |
| llava-onevision-qwen2-72b-ov-hf | 18.010 |
| Qwen2.5-VL-7B-Instruct | 56.250 |
| Qwen2.5-VL-32B-Instruct | 58.962 |
| Qwen2.5-VL-72B-Instruct | 58.962 |
| Qwen3-VL-235B-A22B-Instruct | 58.350 |
| Molmo-7B-D | 61.632 |
| Molmo-7B-O | 63.266 |
| Molmo-72B | 63.832 |
| VisionReasoner-7B | 64.766 |
| Poivre-7B (Ours) | **67.515** |

Table 2: Comparison of two rewards on Point-Bench. The Poivre-7B checkpoint is trained by our PBRS-inspired process reward. The best results are **bolded**.

|  | Success Rate |
| --- | --- |
| Poivre-7B-OutcomeReward | 66.293 |
| Poivre-7B | **67.515** |

margin of 2.7% in success rate. This substantial gain demonstrates the superiority of our training paradigm and establishes a new performance frontier for visual pointing. Relative to Qwen2.5-VL-7B-Instruct, the initialization checkpoint for our RL training, Poivre-7B achieves a remarkable performance gain. Notably, the runner-up, VisionReasoner-7B, is also initialized from the same checkpoint. The superiority of our model arises from incentivizing self-refinement through multi-turn RL, in contrast to VisionReasoner, which employs only single-turn RL. Since VisionReasoner is trained for broader tasks beyond visual pointing, we include additional controlled comparisons in the following experiments. To facilitate further research, we publicly release the model checkpoint.

**Outcome reward vs. PBRS-inspired process reward. (RQ2)** In §4, we introduce a PBRS-inspired process reward as a replacement for the simple outcome reward, and use it to train our Poivre-7B. A natural question arises: what if we simply adopt the straightforward outcome reward? Table 2 compares the two training strategies. While outcome reward alone still enables state-of-the-art performance, our PBRS-inspired process reward yields an additional improvement of about 1.3%. This empirical evidence further validates the effectiveness of the proposed process reward.

**Extrapolation. (RQ3)** As noted earlier, our Poivre-7B is trained with $T = 2$, meaning the model is asked to refine the predicted points only once during RL training. In our experiments, we have so far also fixed $T = 2$ during inference. However, since additional refinement rounds follow the same input format, it is feasible to set $T > 2$ at test time. Table 3 reports the success rates on Point-Bench for different values of $T$. We find that Poivre-7B demonstrates interesting extrapolation ability: when scaling to $T = 3$, beyond what the model has encountered in training, performance further

Table 3: Extrapolation phenomenon on Point-Bench. $T$ is the number of scaling rounds.

|  | Success Rate |
|---|---|
| Poivre-7B (T=1) | 67.108 |
| Poivre-7B (T=2) | 67.515 |
| Poivre-7B (T=3) | 67.617 |
| Poivre-7B (T=4) | 67.617 |
| Poivre-7B (T=5) | 67.617 |

improves. This phenomenon highlights the generalizability of our approach and suggests promising potential for future extensions. While continuing to scale inference time is a possible direction, optimizing this scaling remains an open question for future investigation.

Table 4: Performance of VLMs on the robotics dataset where2place (Yuan et al., 2024a). Asterisk (*) denotes reproduced baselines. Best results are shown in **bold**, while runner-up results are underlined.

|  | Score |
|---|---|
| Qwen-VL | $10.49 \pm 0.77$ |
| LLaVA-NeXT-34B | $15.02 \pm 0.88$ |
| SpaceLLaVA | $11.84 \pm 0.73$ |
| GPT-4o | $29.06 \pm 1.33$ |
| GPT-4o-ICL | $14.46 \pm 6.38$ |
| Molmo-7B-D | $45.00 \pm 0.0$ |
| RoboPoint | $46.77$ $\pm 0.45$ |
| Qwen2.5-VL-7B-Instruct | $40.33* \pm 1.25$ |
| Qwen2.5-VL-32B-Instruct | $43.33* \pm 1.70$ |
| Poivre-7B (Ours) | **49.00** $\pm 0.0$ |

**Generalization to Robotics. (RQ4)** An important application of visual pointing lies in robotics (Cheng et al., 2025). The RoboPoint paper (Yuan et al., 2024a) introduces the where2place dataset for this purpose. Although our Poivre-7B is not explicitly trained on robotics data, it is interesting to examine whether it can generalize directly to this domain. Table 4 reports the performance of Poivre-7B alongside baseline models. For models with publicly available results, we report the performance directly from their original papers. For the Qwen series, we conducted inference ourselves, adopting the same settings as our *Poivre-7B*. The results demonstrate that RL training substantially enhances the pointing ability of the VLM, and this improvement naturally transfers to the robotics setting. A promising direction for future work is to extend our method to robotics training data, and it would also be interesting to evaluate it on real-world tasks, such as grasping (Deshpande et al., 2025).

**Comparison with single-turn RL. (RQ5)** To further validate the necessity of the proposed method, we conduct a comparative experiment with a single-turn RL baseline. For this baseline, which we term VanillaRL, we use the same training configuration on the pointing task but train for only a single turn. Consequently, the model is trained solely with the outcome reward, as our proposed process reward is inapplicable in a single-turn setting. For a fair comparison, we evaluate VanillaRL using the same iterative self-refining procedure at inference. As shown in Figure 3, while this training method improves the initial performance, the ability to self-refine fails to emerge. This result underscores the necessity of our multi-turn training approach for incentivizing this capability. Another noteworthy observation is that our proposed multi-turn RL method yields better results even when the model is evaluated in a single-turn inference setting. While this may appear counter-intuitive at first, it is reasonable because the learned self-refinement capability contributes to a stronger underlying spatial understanding.

**Case study. (RQ6)** Figure 4 presents real-world examples where our Poivre-7B completes the pointing task through refinement. As shown, the model initially points to incorrect coordinates and

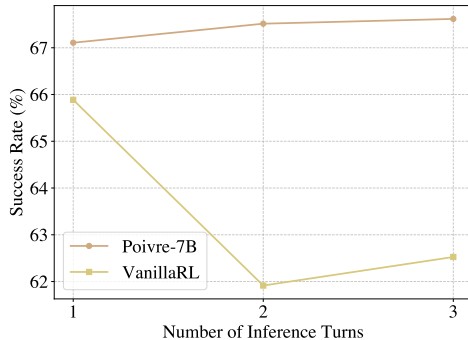

Figure 3: Comparison of Poivre with a baseline trained using vanilla single-turn RL. The single-turn model exhibits severe instability during the self-refining inference process.

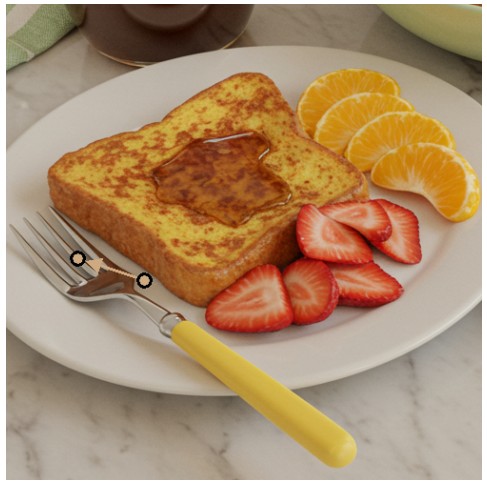
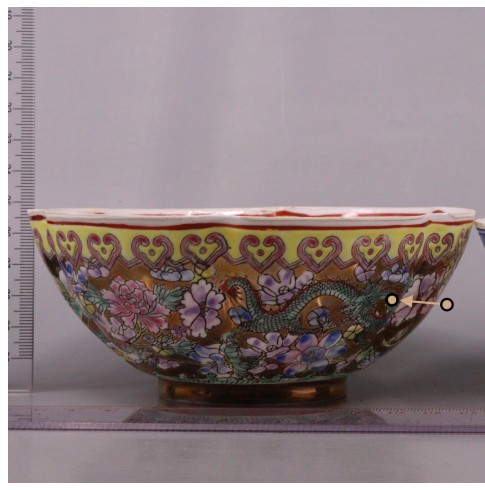

(a) Prompt: Point to the tool used for forking food.          (b) Prompt: Point to the bowl.

Figure 4: Real-world examples where Poivre-7B completes the pointing task through refinement. Brown points denote the visualized coordinates, and arrows indicate the shift from turn 1 to turn 2.

then refines it to accomplish the task. This self-correction behavior serves as a clear demonstration of how our procedure operates in practice.

## 6 CONCLUSION

In this work, we introduce *Point, Visualize, then Refine*, a simple yet effective procedure for visual pointing. By enabling a vision–language model to observe and iteratively refine its own predictions, Poivre makes the pointing process more natural and robust, narrowing the gap between current models and human-like behavior. To incentivize the self-refinement capability of the model, we adopt reinforcement learning with a PBRS-inspired process reward, which is both empirically effective and intuitively appealing. Our resulting model, Poivre-7B, sets a new state of the art on Point-Bench, outperforming both proprietary and large open-source models, and shows strong generalization to robotics tasks without task-specific training. Moreover, we observe an extrapolation effect: Poivre-7B continues to improve when the number of inference rounds exceeds those used during training. Taken together, our results highlight iterative refinement as a powerful paradigm for scaling visual grounding, and we believe this opens promising directions for applying test-time scaling and chain-of-thought to a broader range of multimodal reasoning and control tasks. To support further progress in the community, we release our code, data, and model to the fullest extent possible.

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

## A  PROOF

**Proposition A.1** *The PBRS-inspired process reward can also be viewed as the weighted average of the point rewards across turns, where the initial and the final estimation are most important for usual parameter choices. Formally,*

$$R_{\mathrm{P}}(d_T) = \gamma^{T-1} R_O(d_T) + \sum_{j=1}^{T-1} \gamma^{j-1}(1-\gamma) R_O(d_j), \tag{6}$$

*where $\gamma^{T-1} \geq 1-\gamma \geq \gamma^{j-1}(1-\gamma), \forall j \in [T-1]$, if $T \leq 1 + \frac{\log(1-\gamma)}{\log(\gamma)}$.*

**Proof**  We have

$$R_{\mathrm{P}}(d_T) = R_O(d_1) + \sum_{j=1}^{T-1} \gamma^j (R_O(d_{j+1}) - R_O(d_j)) \tag{7}$$

$$= R_O(d_1) + \sum_{j=1}^{T-1} \gamma^j R_O(d_{j+1}) - \sum_{j=1}^{T-1} \gamma^j R_O(d_j) \tag{8}$$

$$= \sum_{j=1}^{T} \gamma^{j-1} R_O(d_j) - \sum_{j=1}^{T-1} \gamma^j R_O(d_j) \tag{9}$$

$$= \gamma^{T-1} R_O(d_T) + \sum_{j=1}^{T-1} (\gamma^{j-1} - \gamma^j) R_O(d_j) \tag{10}$$

$$= \gamma^{T-1} R_O(d_T) + \sum_{j=1}^{T-1} \gamma^{j-1}(1-\gamma) R_O(d_j). \tag{11}$$

Since $0 < \gamma < 1$, it is trivial to see $\gamma^{T-1} \geq 1-\gamma \geq \gamma^{j-1}(1-\gamma), \forall j \in [T-1]$, if $T \leq 1 + \frac{\log(1-\gamma)}{\log(\gamma)}$. Take the factor ($\gamma = 0.9$) used in our experiments as an example, the condition meets if $T \leq 22$.

## B  FAILURE ANALYSIS

The self-refinement seldom changes correct estimation, the only over-corrected example we can find is presented in Figure 5. We observe the model wants to refine it to reflect where human hold the fork, the estimation falls outside the mask but it is very close.

## C  THE USE OF LARGE LANGUAGE MODELS

We only use large language models for polishing the writing. We understand that we take full responsibility for the contents.

## D  LIMITATIONS AND FUTURE WORK

This paper investigates visual pointing. Specifically, we employ reinforcement learning to encourage the self-refinement capability of VLMs. Our work can be viewed as part of the broader paradigm of test-time scaling and chain-of-thought reasoning. While developing a general-purpose VLM is

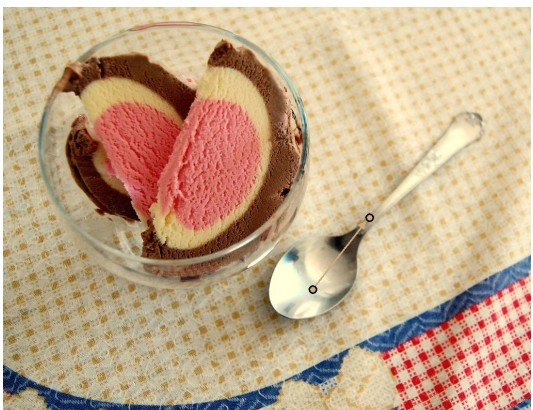

Figure 5: The failure example. Prompt: Point to the tool people use to eat ice cream.

clearly valuable, it is beyond the scope of a single paper to extend our method to all possible vision tasks.

Another promising direction is to leverage the pointing ability for downstream applications, including robotics, assistive technologies, and education. We look forward to exploring these applications in future work.

