# OpenReview forum: "Poivre: Self-Refining Visual Pointing with Reinforcement Learning"
_ICLR.cc/2026/Conference — ICLR 2026 Conference Withdrawn Submission_

### Official Review · Reviewer_cVvG · 2025-10-31

**Soundness:** 3
**Presentation:** 3
**Contribution:** 2
**Rating:** 4
**Confidence:** 3

**Summary:**

This paper proposes Poivre (Point, Visualize, then Refine), a self-refining framework for visual pointing tasks using reinforcement learning (RL). Instead of predicting target coordinates in a single step, Poivre enables a Vision-Language Model (VLM) to iteratively visualize its prediction on the image and refine the coordinates through multiple rounds. The key novelty lies in the PBRS-inspired process reward, which encourages improvement across refinement steps rather than focusing solely on the final outcomes.

**Strengths:**

1.	The paper identifies an important gap in current VLMs: the lack of self-refinement capability. The proposed “Point → Visualize → Refine” loop is simple yet conceptually appealing, like human correction behavior in pointing tasks.
2.	The pipeline and training process (rollout sampling, reward computation) are clearly described, and the paper includes implementation details that make the work easy to replicate.
3.	The adaptation of PBRS to the visual refinement setting is neat and theoretically grounded. It encourages incremental improvement across iterations, which aligns naturally with the proposed iterative reasoning loop.

**Weaknesses:**

1.	While the results are promising, the technical contribution remains incremental. The core ideas, visualizing model predictions and refining them iteratively, are intuitive, and the PBRS-based reward formulation is largely a straightforward adaptation of an established concept (Ng et al., 1999). The work demonstrates solid empirical gains but offers limited novelty.

2.	From Table 3, the improvements across multiple refinement iterations are marginal , which are all within 0.5%. This suggests that after training, the model’s first prediction during testing is already very accurate, making further refinement less impactful. This raises the question of whether the iterative training truly contributes to better refinement behavior, or if the model simply learns to predict correctly in the first step. It would be helpful for the authors to analyze the accuracy gap between the initial prediction and the first refinement during training, to clarify whether the model indeed benefits from iterative supervision.

3.	A similar multi-round refinement mechanism could be implemented in a general supervised setting, where the model iteratively predicts and updates based on the distance between the predicted point and the ground truth. Would such a setup achieve comparable effects without using GRPO? In other words, what is the fundamental difference between performing iterative supervised updates and adopting GRPO in this framework? The paper would benefit from clarifying why reinforcement learning is necessary here, and what unique advantage GRPO provides over conventional multi-step supervised training.

4.	The paper lacks detailed visualization or failure case analysis. It would be useful to show when refinement helps or hurts, and whether the model can meaningfully interpret the marker in complex scenes.

**Questions:**

1.	Have you tried any ablation where the model receives coordinate input directly (without visualization)?
2.	Does the model ever over-correct (move away from the correct target) in refinement steps?
3.	Could a supervised multi-step training baseline achieve similar improvements without RL?
4.	How does the model behave when the initial pointing is already accurate, does refinement still occur?

---

> ### Author Response · Authors · 2025-11-14
> **Author Response**
>
> Dear Reviewer cVvG,
>
> Thank you very much for your detailed review. Your suggestions are very helpful for us to improve our paper and to do research in the future. We would like to take the opportunity of the discussion phase to discuss more details and to potentially seek more guidance from you.
>
> > W1: While the results are promising, the technical contribution remains incremental. The core ideas, visualizing model predictions and refining them iteratively, are intuitive, and the PBRS-based reward formulation is largely a straightforward adaptation of an established concept (Ng et al., 1999). The work demonstrates solid empirical gains but offers limited novelty.
>
> We definitely agree with you that the core ideas, visualizing model predictions and refining them iteratively, are intuitive. And we believe that is a strength of our paper, and the idea is very tempting to try when doing research. For the reward shaping method, we have modified it (we are aware it is a minor one), to make the reward function has two distinct but interesting interpretations. One is the straightforward point then refine reward, the other one is a weighted process reward where the weights actually have intuitive explanations.
>
> > W2: From Table 3, the improvements across multiple refinement iterations are marginal , which are all within 0.5%. This suggests that after training, the model’s first prediction during testing is already very accurate, making further refinement less impactful. This raises the question of whether the iterative training truly contributes to better refinement behavior, or if the model simply learns to predict correctly in the first step. It would be helpful for the authors to analyze the accuracy gap between the initial prediction and the first refinement during training, to clarify whether the model indeed benefits from iterative supervision.
>
> Thank you for your insightful question. We believe the merit of our work does not only lies on the improvement over SOTAs, but also showing the trends of the test-time scaling of self-refining VLMs, and how it is differ from that in the text reasoning domain. In pure text domain, we all aware the success of DeepSeek-R1, where a simple RL can incentive self-refinement ability of LLMs. But we shows that it is different in visual self-refining: In Fig.3, we can see the single-turn RL trained model cannot be steadily scaled. We believe these findings are more interesting that achieving more numerical improvements.
>
> > W3: A similar multi-round refinement mechanism could be implemented in a general supervised setting, where the model iteratively predicts and updates based on the distance between the predicted point and the ground truth. Would such a setup achieve comparable effects without using GRPO? In other words, what is the fundamental difference between performing iterative supervised updates and adopting GRPO in this framework? The paper would benefit from clarifying why reinforcement learning is necessary here, and what unique advantage GRPO provides over conventional multi-step supervised training. Q3: Could a supervised multi-step training baseline achieve similar improvements without RL?
>
> Thank you very much for your suggestion. We agree that multi-step supervised training could also potentially incentives the ability of VLMs. But a major drawback of SFT is that it is hard to collect appropriate data. The best we can get is perhaps the reject sampling, but it is tricky to assess what kinds of data we want to keep. By doing SFT, we also throw away bad cases where could be useful in RL training.
>
> > W4: The paper lacks detailed visualization or failure case analysis. It would be useful to show when refinement helps or hurts, and whether the model can meaningfully interpret the marker in complex scenes. Q2: Does the model ever over-correct (move away from the correct target) in refinement steps?
>
> Thanks for the great suggestion. The self-refinement seldom over-corrects the estimation, we have uploaded the only case in our experiments. The figure can be found in the appendix. One can see that is not a huge mistake, the model only wants to refine it to reflect where human hold the fork, the estimation falls outside the mask but it is very close.
>
> > Q1: Have you tried any ablation where the model receives coordinate input directly (without visualization)?
>
> Thanks for the insightful question, we believe that you mean giving the estimated coordinates back to the model at refinement rounds? That is actually equivalent to multi-rounds text reasoning, as the response of models are also fed back as a part of the context.
>
> > Q4: How does the model behave when the initial pointing is already accurate, does refinement still occur?
>
> In most of cases, we observe the model keep the estimation unchanged. The only overthinking example we can find is added in the appendix.

---

### Official Review · Reviewer_VPKH · 2025-11-01

**Soundness:** 2
**Presentation:** 3
**Contribution:** 3
**Rating:** 4
**Confidence:** 3

**Summary:**

The authors present poivre, a VLM model capable of pointing to images given natural language queries. The model is trained with reinforcement learning, in order to encourage it to refine its predictions. Rather than assigning rewards only for the final state, the authors propose summing the rewards for how much the model improves its prediction at every step; they show this is equivalent to a formulation that emphasizes the first and last prediction the most. Through experiments, they show state-of-the-art results on pointing benchmarks, show that the model's predictions improve with an additional refinement step, and show the model's ability to extrapolate beyond its training regime to a longer horizon.

**Strengths:**

* the paper is well-motivated and clearly written. the figures are helpful towards understanding the method and showcasing qualitative results.
* the composition of ideas seems original, however I am unable to assess this with high confidence as I am not an expert on the literature of visual pointing methods.
* the comparison to single turn RL is interesting! Even with one inference call, the multi-turn objective yields better performance.

**Weaknesses:**

* T is set to 2 during training, which somewhat undercuts the message that the model is refining its responses. It would be very interesting to see how this dynamic plays out over longer sequences, of T=3 and greater.
* As an extension to the above note, it is difficult to access the utility of the PBRS term with just T=2.
* The "exploration" experiments only try T=3 and cite this as evidence of exploration. Please expand the analysis to include longer horizons, T=4, 5, etc, so we can draw more conclusive evidence of extrapolation.
* It would be interesting to compare to additional self-baselines that rely less on visual reasoning and more on text reasoning; for example, just displaying the input image in the original prompt, and asking the model to make text-based prediction outputs, instead of overlaying the prediction onto the original image, as shown currently in Figure 2. This would help tease apart whether **visual** self-feedback is crucial.
* There are sparse citations to several papers in the field related to sampling-based approaches, such as PIVOT (Nasiriany et al.), MOKA (Liu et al.), and Set-of-Mark Prompting (Yang et al.)

**Questions:**

Please see the points raised in the "weaknesses" section.

---

> ### Author Response · Authors · 2025-11-14
> **Author Response**
>
> Dear Reviewer VPKH,
>
> We truly appreciate your insightful review, and we are happy to see your kind words on our motivation and Fig. 3. We also think Fig. 3 is very interesting after we did that experiment. We want to discuss more details with you and humbly seek for more guidance.
>
> > W1: T is set to 2 during training, which somewhat undercuts the message that the model is refining its responses. It would be very interesting to see how this dynamic plays out over longer sequences, of T=3 and greater. W2: As an extension to the above note, it is difficult to access the utility of the PBRS term with just T=2.
>
> Scaling $T$ to longer sequences is definitely an interesting things to do. However, the resources consumption of current RL training framework is massive, and that is even more significant when training VLMs. But we would be thrilled to try that when we have access to more resources in the future.
>
> > W3: The "exploration" experiments only try T=3 and cite this as evidence of exploration. Please expand the analysis to include longer horizons, T=4, 5, etc, so we can draw more conclusive evidence of extrapolation.
>
> Thanks for the insightful question. The performance "converges" after $T=4$, so we did not put it in the original version. We have added it, thank you. We believe it could benefits from larger scale during training, as you suggested in W1 & W2.
>
> > W4: It would be interesting to compare to additional self-baselines that rely less on visual reasoning and more on text reasoning; for example, just displaying the input image in the original prompt, and asking the model to make text-based prediction outputs, instead of overlaying the prediction onto the original image, as shown currently in Figure 2. This would help tease apart whether visual self-feedback is crucial.
>
> Thank you for the insightful question. The VisionReasoner-7B used as our baseline does text reasoning for vision tasks. More importantly, we believe that visual self-feedback makes more sense and is easier to control compared to text reasoning.
>
> > W5: There are sparse citations to several papers in the field related to sampling-based approaches, such as PIVOT (Nasiriany et al.), MOKA (Liu et al.), and Set-of-Mark Prompting (Yang et al.)
>
> Thank you very much for these wonderful works, we have added them as related works.

---

### Official Review · Reviewer_qSiU · 2025-11-01

**Soundness:** 3
**Presentation:** 2
**Contribution:** 2
**Rating:** 4
**Confidence:** 4

**Summary:**

The paper proposes Poivre, a reinforcement learning (RL) framework that enables vision-language models (VLMs) to iteratively refine their pointing predictions through a Point → Visualize → Refine (Poivre) procedure. Rather than predicting a pointing coordinate in one shot, the model visually marks its initial prediction on the image and refines it over multiple rounds. To incentivize this self-refinement, the authors design a potential-based reward shaping (PBRS) mechanism — a “process reward” that encourages progressive improvement across refinement steps, rather than optimizing only for the final outcome.
The resulting model, Poivre-7B, achieves state-of-the-art performance on Point-Bench (67.5% success rate, +2.7% over the best prior open model Molmo-72B) and generalizes to robotics benchmarks (where2place) without domain-specific fine-tuning. The paper includes ablations comparing outcome vs. process rewards, extrapolation to unseen refinement rounds, and a case study illustrating iterative correction.

**Strengths:**

- Clear conceptual motivation: Identifies the one-shot limitation in visual pointing and proposes a self-refining paradigm inspired by human feedback and test-time scaling.

- Principled reward design: The PBRS-inspired process reward is elegant, grounded in RL theory (potential shaping), and practically effective (+1.3% improvement).

- Strong empirical results: New state-of-the-art on Point-Bench, with additional validation on robotics transfer and extrapolation beyond training rounds.

**Weaknesses:**

- Limited conceptual novelty. The idea of self-refinement and iterative improvement is well explored in text reasoning (test-time scaling, visual CoT). The contribution lies mainly in applying this paradigm to visual pointing with straightforward reward shaping, rather than introducing fundamentally new algorithms or representations.
- Modest absolute gains. The improvement (≈2–3%) over strong baselines like VisionReasoner-7B or Molmo-72B, while consistent, is relatively small given the added RL complexity and compute cost (~$2000 training run).
- Reward design tuning unclear. The Gaussian-shaped outcome and PBRS coefficients (σ, γ) are selected heuristically; there’s no sensitivity or stability analysis.
- No human evaluation or qualitative failure analysis. The case studies are anecdotal and do not quantify where refinement helps or fails (e.g., cluttered vs. simple scenes).
- Dependence on GRPO. The method’s improvements might rely heavily on the GRPO setup from prior RL works (DeepSeek-R1, VisionReasoner), rather than intrinsic advantages of Poivre itself.
- No runtime or inference cost discussion. Iterative refinement increases inference time linearly with the number of turns; this tradeoff is not analyzed quantitatively.

**Questions:**

- How sensitive are results to the discount factor (γ) and σ in the process reward? Could these parameters significantly alter performance?

- How many refinement rounds before diminishing returns or overfitting occur (beyond T=3)?

- Can Poivre be extended to spatially dense tasks (e.g., segmentation, affordance maps), not just 2D coordinate regression?

- Did the authors attempt to combine supervised fine-tuning with RLHF/DPO before GRPO?

- What is the inference-time latency cost (in seconds per refinement step) compared to one-shot baselines?

**Details Of Ethics Concerns:**

No notable ethical issues. The work relies on public datasets (Pixmo, Point-Bench, RoboPoint). Potential concerns include model misuse for surveillance or dataset licensing, but these are minimal and standard for vision-language RL research.

---

> ### Author Response · Authors · 2025-11-14
> **Author Response (1/2)**
>
> Dear Reviewer qSiU,
>
> Thank you very much for your insightful review, they are very useful for us to further improve our paper. And we are also very glad to receive your recognition on our motivation and results. During the discussion phase, we would like to take this opportunity to provide additional details of our work and further seek your valuable guidance. We hope that the clarifications below can better address your concerns and help improve the overall quality of the paper.
>
> > W1: Limited conceptual novelty. The idea of self-refinement and iterative improvement is well explored in text reasoning (test-time scaling, visual CoT). The contribution lies mainly in applying this paradigm to visual pointing with straightforward reward shaping, rather than introducing fundamentally new algorithms or representations.
>
> We agree that self-refinement methods are successful in the field of text reasoning, and that is precisely the motivation of our method and the reason for the recent surge of the "think-with-image" researches. Despite its popularity in pure text domain, we believe that it is still worth it to studies such methods on visual tasks like pointing. As you accurately point out in S1, the motivation is clearly from human behavior, and that makes it natural to implement such ideas. Introducing fundamentally new algorithms or representations are indeed important works, but we believe that does not means it is not worth to try simple and intuitive approaches.
>
> > W2: Modest absolute gains. The improvement (≈2–3%) over strong baselines like VisionReasoner-7B or Molmo-72B, while consistent, is relatively small given the added RL complexity and compute cost (~$2000 training run).
>
> Thank you for your comment. We would like to kindly highlight that our Poivre-7B is trained from the Qwen2.5-VL-7B-Instruct checkpoint, and we use the ~$2000 compute cost to make it surpass Molmo-72B and proprietary models such as Gemini-2.5-pro. Although the exact figure is not accessible, but we believe it is safe to say that the post-training cost of Gemini-2.5-Pro should be much more expensive than ours. We are open to more discussion on this topic.
>
> > W3: Reward design tuning unclear. The Gaussian-shaped outcome and PBRS coefficients (σ, γ) are selected heuristically; there’s no sensitivity or stability analysis. Q1: How sensitive are results to the discount factor (γ) and σ in the process reward? Could these parameters significantly alter performance?
>
> Thank you for your suggestions. As stated in line 237, we use the Gaussian-shaped reward function since it naturally captures diminishing marginal improvement when the grounding is more accurate (i.e., $d_T$ is small). We can share an early experimental result, where we compare the Gaussian-shaped reward to a coarser reward, $R(d_T)=1/1+d_T$. Note the results are single-turn RL rather than our Poivre. This further motivate us to adopt such intuitive reward function. In the moment, we are not able to conduct more sensitivity analysis due to the high training cost, but we believe such choice does not affect the merit of our work.
>
> |    | Success Rate |
> | -------- | ------- |
> | Gaussian  | 65.886   |
> | Inverse | 64.664  |
>
> > W4: No human evaluation or qualitative failure analysis. The case studies are anecdotal and do not quantify where refinement helps or fails (e.g., cluttered vs. simple scenes).
>
> Thanks for the great suggestion. The self-refinement seldom over-corrects the estimation, we have uploaded the only case in our experiments. The figure can be found in the appendix. One can see that is not a huge mistake, the model only wants to refine it to reflect where human hold the fork, the estimation falls outside the mask but it is very close.

---

> ### Author Response · Authors · 2025-11-14
> **Author Response (2/2)**
>
> > W5: Dependence on GRPO. The method’s improvements might rely heavily on the GRPO setup from prior RL works (DeepSeek-R1, VisionReasoner), rather than intrinsic advantages of Poivre itself.
>
> Thank you for the insightful question. We want to kindly highlight Fig. 3, where we compare the VanillaRL (only use GRPO) and our method (two-turn RL to incentive self-refinement). We observe that our method shows a steady test-time scaling curve (though it is not astonishing due to the scale), but the curve of GRPO-only drops when $T$ increases. While GRPO itself can improve the performance as well, the most important message we want to convey is that it does not automatically incentive self-refinement unless we explicitly design it.
>
> > W6: No runtime or inference cost discussion. Iterative refinement increases inference time linearly with the number of turns; this tradeoff is not analyzed quantitatively. Q5:  What is the inference-time latency cost (in seconds per refinement step) compared to one-shot baselines?
>
>
> Thank you for pointing this out. It is true that the inference time will linearly increase with the number of turns, we provide the average runtime here.
>
> |  $T$  | Ave. Time per query |
> | -------- | ------- |
> | 1  | 2.05s  |
> | 2 | 4.10s |
> |3|6.37s|
>
> > Q2: How many refinement rounds before diminishing returns or overfitting occur (beyond T=3)?
>
> Thanks for the insightful question. The performance "converges" after $T=4$, so we did not put it in the original version. We have added it, thank you.
>
> > Q3: Can Poivre be extended to spatially dense tasks (e.g., segmentation, affordance maps), not just 2D coordinate regression?
>
> Thank you for your insightful question. We believe visual self-refinement methods are very promising on these task, and we are currently doing multiple projects toward these application.
>
> > Q4: Did the authors attempt to combine supervised fine-tuning with RLHF/DPO before GRPO?
>
> This is indeed an important question to discuss, thank you. As our Poivre-7B is trained from Qwen2.5-VL-7B-Instruct, we want to kindly highlight Page 6 of the arXiv version of Qwen2.5-VL Technical Report, where the PixMo datasets are cited. That paragraph confirms that the SFT you suggested is already conducted by the Qwen team, so we have not do it the second time in our experiment. Do you suggest we include this discussion into our paper?

---

### Official Review · Reviewer_VxU2 · 2025-11-03

**Soundness:** 1
**Presentation:** 1
**Contribution:** 2
**Rating:** 2
**Confidence:** 3

**Summary:**

Visual pointing, which aims to localize a target by predicting its coordinates on an image, has emerged as an important problem in the realm of vision–language models (VLMs). Despite its broad applicability, recent benchmarks show that current VLMs still fall far behind human performance on this task. A key limitation is that VLMs are typically required to complete the pointing task in a single step, akin to asking humans to point at an object without seeing their own fingers. To address this issue, we propose a simple yet effective self-refining procedure: Point, Visualize, then Refine (Poivre). This procedure enables a VLM to first mark its estimated point, then iteratively refine the coordinates if necessary. Inspired by advances of reasoning models in the natural language domain, we employ reinforcement learning (RL) to incentivize this self-refining ability. For the RL training, we design a neat process reward that is not only empirically effective but also grounded in appealing theoretical properties. Our trained model, Poivre-7B, sets a new state of the art on Point-Bench, outperforming both proprietary models such as Gemini-2.5-Pro and strong open-source models such as Molmo-72B by over 3%. To support future research, we release our training and inference code, dataset, and the Poivre-7B checkpoint.

**Strengths:**

1) Extensive experiments show the effectiveness of the proposed method.

**Weaknesses:**

1) The writing should be improved.
2) The motivation should be further highlighted.
3) The paper organization is poor.
4) Figures in the paper should be enhanced.

**Questions:**

1) The writing should be improved.
2) The motivation should be further highlighted.
3) The paper organization is poor.
4) Figures in the paper should be enhanced.

---

### Author Response · Authors · 2025-11-22

Dear Reviewers,

Thank you again for your insightful and constructive review. It has been a week since we submitted our response and the revised version of our paper. We would greatly appreciate it if you could take a moment to review the updates and provide any additional discussions. Your suggestions have been extremely helpful to us and to the community.

We also wish you the best of luck with any submissions of your own.

---

### Note · Authors · 2026-01-05

I have read and agree with the venue's withdrawal policy on behalf of myself and my co-authors.